# Cervical kinematic change after posterior full-endoscopic cervical foraminotomy for disc herniation or foraminal stenosis

Seungyoon Paik[1☯], Yunhee Choi[2☯], Chun Kee Chung[1,3,4,5], Young Il Won[6], Sung Bae Park[3,4,7], Seung Heon Yang[3,8], Chang-Hyun Lee[3,4], John Min Rhee[9], Kyoung-Tae Kim[10,11], Chi Heon Kim[3,4,12]*

1 Department of Medicine, Seoul National University College of Medicine, Seoul, Republic of Korea, 2 Division of Medical Statistics, Medical Research Collaborating Center, Seoul National University Hospital, Seoul, Republic of Korea, 3 Department of Neurosurgery, Seoul National University College of Medicine, Seoul, Republic of Korea, 4 Department of Neurosurgery, Seoul National University Hospital, Seoul, Republic of Korea, 5 Department of Brain and Cognitive Sciences, Seoul National University, Seoul, Republic of Korea, 6 Department of Neurosurgery, Chungnam National University Sejong Hospital, Sejong, Republic of Korea, 7 Department of Neurosurgery, Seoul National University Boramae Hospital, Seoul, Republic of Korea, 8 Department of Neurosurgery, Seoul National University Bundang Hospital, Seongnam-si, Gyeonggi-do, Republic of Korea, 9 Department of Orthopaedic Surgery, Emory University School of Medicine, Atlanta, Georgia, United States of America, 10 Department of Neurosurgery, Kyungpook National University Hospital, Daegu, Republic of Korea, 11 Department of Neurosurgery, School of Medicine, Kyungpook National University, Daegu, Republic of Korea, 12 Department of Medical Device Development, Seoul National University College of Medicine, Seoul, Republic of Korea

☯ These authors contributed equally to this work.
* chiheon1@snu.ac.kr

**Data Availability Statement:** All relevant data are within its Supporting Information files.

## Abstract

### Objective

Posterior full-endoscopic cervical foraminotomy (PECF) is one of minimally invasive surgical techniques for cervical radiculopathy. Because of minimal disruption of posterior cervical structures, such as facet joint, cervical kinematics was minimally changed. However, a larger resection of facet joint is required for cervical foraminal stenosis (FS) than disc herniation (DH). The objective was to compare the cervical kinematics between patients with FS and DH after PECF.

### Methods

Consecutive 52 patients (DH, 34 vs. FS, 18) who underwent PECF for single-level radiculopathy were retrospectively reviewed. Clinical parameters (neck disability index, neck pain and arm pain), and segmental, cervical and global radiological parameters were compared at postoperative 3, 6, and 12 months, and yearly thereafter. A linear mixed-effect model was used to assess interactions between groups and time. Any occurrence of significant pain during follow-up was recorded during a mean follow-up period of 45.5 months (range 24–113 months).

### Results

Clinical parameters improved after PECF, with no significant differences between groups. Recurrent pain occurred in 6 patients and surgery (PECF, anterior discectomy and fusion)

**Funding:** This work was supported by the New Faculty Startup Fund from Seoul National University. This study was supported by grant no. 04-2021-0540 from Seoul National University Hospital research fund. This study was supported by Doosan Yonkang foundation (800-20210527). There was no additional external funding received for this study. The funders had no role in study design, data collection and analysis, decision to publish, or preparation of the manuscript.

**Competing interests:** I have read the journal's policy and the authors of this manuscript have the following competing interests: The corresponding author (CHK) is a consultant of RIWOspine GmBH. All the other authors declare that they have no conflicts of interest concerning the materials/methods used in this study or the findings described in this paper. No benefits in any form have been or will be received from any commercial party related directly or indirectly to the subject of this manuscript. This does not alter our adherence to PLOS ONE policies on sharing data and materials.

was performed in 2 patients. Pain-free survival rate was 91% for DH and 83% for FS, with no significant difference between the groups (P = 0.29). Radiological changes were not different between groups (P > 0.05). Segmental neutral and extension curvature became more lordotic. Cervical curvature became more lordotic on neutral and extension X-rays, and the range of cervical motion increased. The mismatch between T1-slope and cervical curvature decreased. Disc height did not change, but the index level showed degeneration at postoperative 2 years.

## Conclusion

Clinical and radiological outcomes after PECF were not different between DH and FS patients and kinematics were significantly improved. These findings may be informative in a shared decision-making process.

## Introduction

In patients with radiculopathy due to foraminal disc herniation or stenosis, surgery is recommended when non-surgical treatment is not effective [1–4]. Currently, surgical options include anterior cervical discectomy fusion (ACDF), artificial disc replacement, and posterior microforaminotomy [4–7]. Although ACDF and artificial disc replacement are well-established and popular surgical methods, surgery without instrumentation while preserving cervical motion would be a good alternative to those surgical methods [8, 9]. Clinical outcomes were not found to be different between posterior foraminotomy and ACDF during a 5-year follow-up [8]. However, disruption of spinal kinematics and subsequent re-operation are concerns after foraminotomy [10, 11]. Although a systematic review in 2016 showed a similar reoperation rate between ACDF and posterior foraminotomy (4% vs. 6%) [12], a study conducted using data from the national Swedish spine register (Swespine) showed that the reoperation rate was significantly higher after posterior foraminotomy than after ACDF (6% vs. 1%, P < 0.01) [8]. Nonetheless, there are many potential advantages of posterior foraminotomy, such as the ability to achieve similar clinical outcomes with lower medical costs, and a lower incidence of adjacent segment disease than is the case of ACDF, may also be an attractive advantage of posterior foraminotomy [12, 13]. However, concerns remain regarding the unfavorable consequences of partial facetectomy, such as progression of cervical kyphosis, loss of cervical lordosis, and re-operation [10]. ACDF continues to be preferred as a surgical option over foraminotomy, as shown in the Swespine study, in which 3721 of 4368 (85%) patients underwent ACDF, while 647 of 4368 (15%) patients underwent posterior foraminotomy [8]. Recently, posterior full-endoscopic cervical foraminotomy (PECF) emerged as a minimally invasive surgical technique, and its influence on changes in cervical kinematics may not be as significant as that of open surgery [14–18]. A recent systematic review in 2019 showed a similar reoperation rate (3.9% vs. 6.9%) and complication rate (7.8% vs. 4%) between ACDF and minimally invasive posterior cervical foraminotomy [6]. However, some problems still remain. Despite the minimally invasive nature of PECF, injury of the facet joint and musculature may increase the likelihood of progression to cervical kyphosis or loss of lordosis in patients with foraminal stenosis (FS) compared to patients with disc herniation (DH), because more extensive removal of the facet joint is necessary in patients with FS than in those with DH [9, 10,

19]. Therefore, the present study was planned to compare cervical kinematics between patients with DH and those with FS after PECF.

## Materials and methods

### Patients

After obtaining permission from the institutional review board, the prospectively collected medical records of patients who underwent PECF between June 2010 and February 2018 were retrospectively reviewed. The prospective collection of clinical and radiological data and retrospective review of the data were approved by the Institutional Review Board (No. H-0507-509-153 and 2101-080-118, respectively). Informed consent was obtained from all individual participants included in this study for the prospective collection of data. However, the requirement for informed consent was waived for the retrospective review because this study involved no more than minimal risk and would not adversely affect the rights and welfare of the patients.

This study included patients 1) with single-level unilateral radiculopathy due to cervical DH or FS, 2) a positive Spurling's test, 3) disc space narrowing not more than 50%, [20] 4) complete preoperative clinical and radiological data, and 5) postoperative follow-up for more than 2 years. Patients with 1) previous cervical or lumbar spinal surgery, 2) malignancy, inflammatory joint disease, trauma, psychiatric disease and neuromuscular disease, and 3) ossification of the posterior longitudinal ligament were excluded [15, 19, 21]. For DH, a foraminal soft disc herniation was confirmed with computed tomography (CT) scan and magnetic resonance imaging (MRI) without any bony foraminal stenosis. All patients with bony foraminal stenosis confirmed by CT and MRI were classified as FS. Finally, 52 patients (DH, 34 vs. FS, 18) were included in this study.

### Surgical techniques

The surgical techniques of PECF were similar to those previously reported [14–17, 22]. PECF was performed in a prone position under general anesthesia. The surgical level was identified with C-arm fluoroscopy and a skin incision of 8 mm was made at skin above the "V-point," which is formed by the lamina, descending facet, and ascending facet [14–17]. The obturator (6.9 mm outer diameter), working tube (8.0 mm outer diameter) and endoscope (4.1-mm working channel, Vertebris®, Richard Wolf GmbH, Knittlingen, Germany) were sequentially introduced through the skin incision [14–17]. Laminectomy and facetectomy were performed using an endoscopic drill under direct visualization. The size of the bone drilling was dependent on the size and location of the herniated disc material and the extent of stenosis, and it was usually within a radius of 3–4 mm around the V-point for soft disc herniations and 5–6 mm for foraminal stenosis [14–17, 19]. The herniated disc was removed through the axilla or shoulder of the nerve root. Excluding the size of bone drilling and removal of disc material, there was no other difference between the surgical procedures of DH and FS. Decompression and free movement of the nerve root were conformed at the shoulder/axilla and superolateral/inferolateral corner of the nerve root. A closed suction drain was inserted through the working tube and the skin was closed. Patients were encouraged to walk on the day of surgery without a neck brace and discharged the next day without limitations of neck motion [15, 19].

### Clinical evaluation

Information on weight, height, occupation, smoking status, and diabetes was collected during a preoperative interview with a nurse. A questionnaire including the Neck Disability Index

(NDI, out of 50) [23], numerical rating scales of neck pain (Neck-NRS, out of 10) and arm pain (Arm-NRS, out of 10) were filled out by every patient preoperatively. After surgery, patients were scheduled to visit the clinic at 1, 3, 6, 12 months and yearly thereafter, and filled out the same questionnaire at every visit. Occupations were categorized into three categories according to occupational activity (OA): high OA, intermediate OA, and low OA [24]. Those data were prospectively recorded in the electronic medical records system of the hospital. During follow-up, re-appearance of significant pain was recorded and it was defined as an "adverse event" in this study. The pain was first managed by medication for 1–2 weeks and interventions such as epidural injection were performed by pain physicians if medication did not work. When those measures failed to control pain, surgery was recommended. Patients were followed up for 45.5 ± 20.6 months (range, 24–113 months).

## Radiological evaluation

Preoperatively, patients underwent MRI, CT and plain X-rays, which included cervical lateral neutral, flexion, and extension X-rays and whole-spine anterior-posterior and lateral X-rays. At the follow-up clinic visits, X-rays were taken at 3, 6, and 12 months and yearly thereafter. All X-rays were taken using the same protocol: The patients were asked to stand and look straight ahead for the neutral-position and whole-spine radiographs, and to flex and extend their neck to the extent they could tolerate for the flexion and extension X-rays [14, 15]. The radiological parameters were evaluated in three aspects: local, regional, and global ones (Fig 1). Locally, the index level segmental neutral angle (SA-N), segmental flexion angle (SA-F), segmental extension angle (SA-E), segmental range of motion (S-ROM), anterior disc height (aH), posterior disc height (pH), and cervical degenerative index (CDI) at the surgical level were assessed [20]. The magnification ratio was assessed by measuring the anterior-posterior lengths of cranial vertebral bodies on plain X-rays and computed tomography (CT), and the ratio was used to calculate the actual aH and pH. Regionally, the C2 to C7 sagittal vertical axis (C27-SVA), T1 slope (T1S), cervical neural curvature from C2-7 (CA-N), cervical flexion curvature (CA-F), cervical extension curvature (CA-E), difference between T1 slope and CA-N (T1S-CA), and cervical range of motion (C-ROM) were evaluated. Globally, the C7 sagittal vertical axis (C7-SVA) and T1 pelvic angle (TPA) were evaluated (Fig 1). The measurements and the analysis were performed on 150% magnified images using measuring tools in the institution's picture archiving and communication system (Marosis, version 5483, Infinitt Healthcare, Seoul, Korea), which was run in a Microsoft Windows environment (Microsoft Corp., Redmond, WA, USA) [25]. All of the above parameters were measured by an independent researcher (Blinded for review). The methods of measurement are described in detail in Fig 1. The CDI was scored 0–3 for each of the following 4 categories: narrowing of the disc space, presence of bony sclerosis, osteophytes, and olisthesis [20]. A CDI of 0 means no degeneration, while a score of 12 indicates severe degeneration [20].

## Statistical analysis

The patients were divided into two groups: DH (n = 34) and FS (n = 18), and the variables were summarized using mean (standard deviation) or frequency (proportion). After performing the normality tests, these variables were compared between groups using the t-test or chi-square test, as appropriate. The Kaplan-Meier method was used to assess the event-free survival time, and the log-rank test was used to make between-group comparisons. A linear mixed-effect model was used to assess the changes in clinical parameters (NDI, Neck-NRS, and Arm-NRS) and radiological parameters. The fixed effects were group, time, the interaction between group and time, age, diabetes, smoking, body mass index (BMI, kg/m$^2$), and sex. The

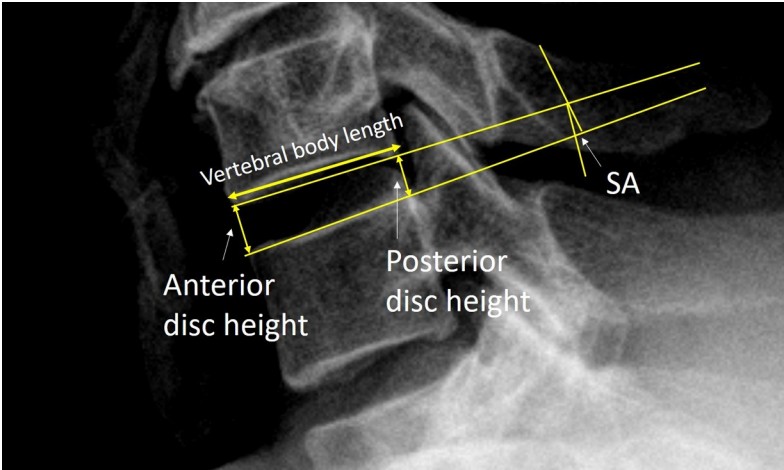

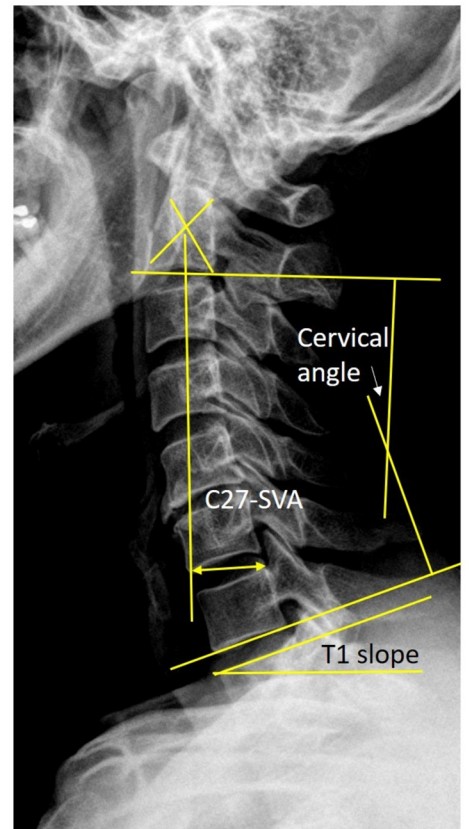

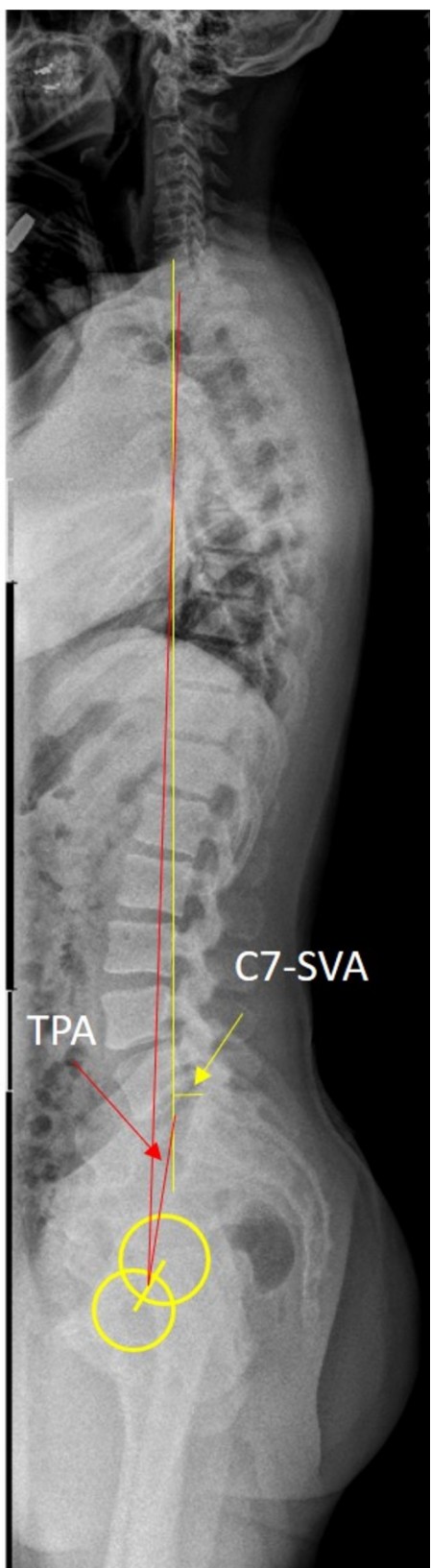

**Fig 1. Radiological measurements.** a. The Cobb method was used to measure the segmental angle between the superior and inferior endplates of the index disc in neutral (SA-N), flexion (SA-F) and extension (SA-E) X-rays. Segmental range of motion (sROM) was calculated by subtracting SA-E from SA-F. The anterior and posterior heights of discs were measured on X-rays. The magnification ratio was assessed by measuring the anterior-posterior lengths of cranial vertebral bodies on plain X-rays and computed tomography, and the ratio was used to calculate the actual anterior disc height (aH) and posterior disc height (pH). b. Cervical lordosis was measured using the Cobb method from the inferior endplate of C2 to the inferior endplate of C7 in neutral (CA-N), flexion (CA-F) and extension (CA-E) X-rays. Cervical range of motion (C-ROM) was calculated by subtracting CA-E from CA-F. The sagittal vertical axis was defined as the horizontal length between the vertical plumb line from the center of the C2 body to the posterosuperior corner of C7 (C27-SVA). T1 slope (TS) was measured between the horizontal line and the extension line along the superior endplate of T1. c. The sagittal vertical axis was measured from the center of the C7 vertebral body to the posterosuperior corner of the S1 vertebral body (C7-SVA). The T1 pelvic angle (TPA) was defined as the angle between 2 lines from the line from the centroid of the bicoxofemoral axis to the centroid of T1 and to the middle of the S1 superior endplate.

random effect was a random intercept. The interaction between group and time was tested with a 0.01 significance level to control the rate of false-positive interactions due to the number of parameters tested. A post hoc analysis using the Bonferroni method was planned for significant time effects; differences in clinical and radiological parameters between before and after the operation, and between 3 months and the other time points after the operation. Therefore, the significance levels for the post-hoc test were 0.006 and 0.007 for clinical and radiological parameters, respectively. All of the statistical analyses were performed using SAS® version 9.4 (SAS Institute Inc., Cary, NC, USA).

## Results

The characteristics of the patients are described in Table 1. The mean age of the DH group was 47.7 years and that of the FS group was 46.3 years (P = 0.28). C6-7 was the most common surgical level, followed by C5-6, C7-T1, and C4-5; the distribution was not statistically different between groups (P = 0.89). BMI was similar between the groups (P = 0.09) and the proportion of patients with smoking or diabetes mellitus was not significantly different between the groups (P = 1.00 and 0.11, respectively). Most patients (43/52, 83%) worked in jobs with intermediate OA, and the distribution was not significantly different between the groups (P = 0.60). There were no significant differences in clinical and radiological parameters (P > 0.05), except for NRS-Arm, which was significantly higher in the DH patients than in the FS patients (P = 0.002), and T1S, which was higher in the FS patients than in the DH patients (P = 0.01).

### Clinical outcomes

Table 2 and Fig 2 show the observed mean values (standard deviation) and the mean values with 95% confidence intervals (CIs) adjusted by OA, BMI, age and sex, respectively. The adjusted mean of the NDI and Neck-NRS were significantly different across the time points (P < 0.001) in each group and the adjusted mean values of NDI and Neck-NRS were lower in DH patients than in FS patients by 1.51 (95% CI: -0.62 to 3.65; P = 0.16) and 0.25 (95% CI: -0.33 to 0.84; P = 0.39), respectively, but these differences were not statistically significant. The NDI decreased significantly at postoperative 1 month (P <0.006) and was lowest at postoperative 3 and 4 years (Table 2 and Fig 2). Neck-NRS significantly decreased at postoperative 1 month and there was no further change during the follow-up period (Table 2 and Fig 2). The interaction between group and time was significant for Arm-NRS (P = 0.002), which may have been caused by different patterns of changes within the two groups, as the Arm-NRS was higher in the DH group before the operation, but slightly higher in the FS group after the operation. The adjusted mean of Arm-NRS significantly decreased after the operation (adjusted P < 0.006) in each group, but there was no further significant change during the follow-up period.

**Table 1. Characteristics of patients.**

| | | Total (n = 52) | Disc herniation (n = 34) | Foraminal stenosis (n = 18) | P-value |
|---|---|---|---|---|---|
| Age, mean (SD) | | 47.7 (13.1) | 46.3 (11.1) | 50.4 (16.1) | 0.28 |
| Sex (female) | | 16 (31%) | 13 (38%) | 3 (17%) | 0.11 |
| Side (right) | | 19 (37%) | 12 (35%) | 7 (39%) | 0.8 |
| Level | C4-5 | 5 (10%) | 4 (12%) | 1 (6%) | 0.89* |
| | C5-6 | 17 (33%) | 10 (29%) | 7 (39%) | |
| | C6-7 | 24 (46%) | 16 (47%) | 8 (44%) | |
| | C7-T1 | 6 (12%) | 4 (12%) | 2 (11%) | |
| BMI (kg/m$^2$) | | 24.8 (3.3) | 24.2 (3.2) | 25.8 (3.1) | 0.09 |
| Smoking (yes) | | 9 (17%) | 6 (18%) | 3 (17%) | >0.99* |
| Diabetes mellitus (yes) | | 4 (8%) | 1 (3%) | 3 (17%) | 0.11* |
| Occupational activity | High | 2 (4%) | 2 (6%) | 0 (0%) | 0.60* |
| | Intermediate | 43 (83%) | 27 (80%) | 16 (89%) | |
| | Low | 7 (14%) | 5 (15%) | 2 (11%) | |
| NDI, mean (SD) | | 23.1 (7.1) | 24.2 (7.9) | 21 (5) | 0.08 |
| Neck-NRS | | 6.3 (2.1) | 6.7 (2.1) | 5.6 (2.1) | 0.08 |
| Arm-NRS | | 7 (2.1) | 7.6 (2) | 5.8 (1.8) | < 0.01 |
| C7-SVA | | -13.1 (21.9) | -16.6 (21.4) | -6.4 (21.8) | 0.11 |
| T1tilt | | 5.8 (2.6) | 6.3 (2.5) | 4.9 (2.5) | 0.07 |
| TPA | | 15.5 (5.2) | 15.5 (5.9) | 15.5 (3.6) | 0.98 |
| C27-SVA | | 24.4 (14.8) | 21.7 (11.7) | 29.4 (18.7) | 0.13 |
| T1S | | 20.7 (7.6) | 18.7 (6.5) | 24.6 (8.3) | 0.01 |
| T1S-CA | | 16.3 (11.5) | 16.3 (9.1) | 16.5 (15.3) | 0.95 |
| CA-N | | -4.4 (12) | -2.4 (10.3) | -8.1 (14.2) | 0.10 |
| CA-F | | 19.5 (12.2) | 20.1 (12.6) | 18.3 (11.8) | 0.62 |
| CA-E | | -15.4 (15.5) | -15 (12.6) | -16.2 (20.3) | 0.82 |
| C-ROM | | 34.9 (20.8) | 35.1 (20.5) | 34.5 (22.1) | 0.92 |
| SA-N | | 0.4 (4.9) | 0.9 (5.5) | -0.4 (3.4) | 0.32 |
| SA-F | | 6.8 (6.1) | 7.2 (6.3) | 6.2 (5.7) | 0.57 |
| SA-E | | -2.8 (5.5) | -3.1 (5.6) | -2.3 (5.5) | 0.59 |
| S-ROM | | 9.6 (6.6) | 10.3 (6.7) | 8.4 (6.3) | 0.33 |
| CDI | | 1.3 (1.3) | 1.1 (1.2) | 1.6 (1.4) | 0.19 |
| aH | | 3.0 (1.3) | 3.1 (1.2) | 2.9 (1.5) | 0.54 |
| pH | | 2.5 (1.1) | 2.6 (0.9) | 2.4 (1.3) | 0.38 |

Abbreviations: BMI, body mass index; NDI, neck disability index; Neck-NRS, numerical rating scale scores for neck pain; Arm-NRS, numerical rating scale for arm pain; C7-SVA, C7 sagittal vertical axis; TPA, T1 pelvic angle; C27-SVA, C2 to C7 sagittal vertical axis; T1S, T1 slope; CA-N, cervical neural curvature from C2-7; CA-F, cervical flexion curvature; CA-E, cervical extension curvature; T1S-CA, difference between T1 slope and CA-N; C-ROM, cervical range of motion; SA-N, index level segmental neutral angle; SA-F, segmental flexion angle; SA-E, segmental extension angle; S-ROM, segmental range of motion; aH, anterior disc height; pH, posterior disc height; CDI, cervical degenerative index.

*Fisher's exact test

The patients in this study did not experience any direct surgery-related complications such as nerve palsy, dysesthesia, or dural tear [26]. Adverse events occurred in 6 patients (12%) during follow-up. In the DH group, 3 patients experienced events at 15 months, 36 months, and 43 months and the events were controlled by an epidural injection at the index level, nucleoplasty at the index level, and epidural injections at the below level, respectively. In the FS group, 3 patients experienced events at 27 months, 30 months, and 36 months and the events were controlled by PECF at the below level, epidural injection at the index level, and anterior cervical

**Table 2. Clinical outcomes: Adjusted mean (standard deviation) [*].**

|  |  | Pre | 1 mo | 3 mo | 6 mo | 1 yr | 2 yr | 3 yr | 4 yr | 5 yr |
|---|---|---|---|---|---|---|---|---|---|---|
| NDI Mean (SD) | DH | 24.2 (7.9) | 9.1 (5.9) | 6 (5.2) | 4.9 (4.2) | 4.9 (5.5) | 3.1 (4.9) | 2.5 (6.1) | 2.4 (4.5) | 5.7 (9.8) |
|  | FS | 21 (5) | 10.6 (5.8) | 7.7 (5.3) | 6.9 (5.7) | 6.3 (4.5) | 7.1 (5.9) | 4.2 (6.9) | 3.5 (7.2) | 6 (8.2) |
| Neck-NRS | DH | 6.7 (2.1) | 2.5 (1.8) | 1.3 (1.4) | 1.6 (1.7) | 1.6 (2) | 0.7 (1.4) | 0.8 (1.6) | 1.3 (1.9) | 2.5 (3.3) |
|  | FS | 5.6 (2.1) | 2.1 (1.6) | 1.7 (1.6) | 1.1 (1.3) | 1.2 (1.4) | 1.9 (2) | 2.2 (3.2) | 2.9 (3.8) | 3 (3.6) |
| Arm-NRS | DH | 7.6 (2) | 2.3 (2) | 1.6 (1.6) | 1.8 (1.8) | 1.8 (2) | 1.4 (1.8) | 1 (2.3) | 1.8 (2.9) | 2.3 (3.2) |
|  | FS | 5.8 (1.8) | 3.3 (2.8) | 1.9 (1.5) | 2 (1.8) | 2.1 (2.1) | 1.4 (2.1) | 2.9 (3.7) | 1.4 (3.1) | 2 (3.5) |

[*]Adjusted mean values by age, sex, BMI, occupational activity, and preoperative clinical outcomes were calculated from the mixed-effect model

[†]Adjusted mean for time (95% confidence interval)

[‡]Statistically significant at the 0.05 level.

Abbreviations: NDI, neck disability index; neck-NRS, numerical rating scale scores for neck pain; Arm-NRS, numerical rating scale for arm pain; DH, disc herniation (n = 34); FS, foraminal stenosis (n = 18).

discectomy and fusion at the index level, respectively. Overall, the 5-year event-free survival rate was 80% (95% CI: 66%–95%), and it was not different between groups (P = 0.29) (Fig 3).

## Radiological outcomes

Radiological parameters were presented in three aspects: locally, regionally, and globally (Table 3). None of the radiological outcomes showed significant interactions between groups and across time points (P > 0.01). Locally, SA-N, SA-F, SA-E, and CDI significantly changed after the operation (P < 0.007), while a significant change was not observed in S-ROM, aH, and pH across time points (Fig 4A–4D). The CDI showed that degeneration had progressed at postoperative 2 years (P < 0.007) (Fig 4D). CDI increased in 6 patients (18%) in the DH group and 5 patients (28%) in the FS group (P = 0.48). The 5-year degeneration-free survival rates were 70% (95% CI: 50%–90%) for the DH group and 60% (95% CI: 30%–90%) for the FS group, without a significant difference between groups (P = 0.32).

Regionally, C27-SVA significantly decreased at postoperative 3 months and maintained throughout follow-up period (Fig 4E). Regional parameters, C27-SVA, CA-N, CA-E, C-ROM and T1S-CA mismatch, showed a significant change at postoperative 3 months (P < 0.007), but there was no further change thereafter (P > 0.007) (Table 3 and Fig 4E–4I). However, CA-F did not show a significant change across time points (p > 0.007).

Globally, C7-SVA significantly changed after surgery and the change was maintained throughout the follow-up period (Fig 4J). TPA did not show any significant change (Table 3).

## Case

A 42-year-old female patient visited with neck and right arm pain (NDI 35/50, Neck-NRS 10/10, and Arm-NRS 10/10). Preoperative T2-weighted magnetic resonance imaging showed foraminal disc herniation at left C6-7. Cervical radiographs showed kyphotic cervical and segmental curvature (SA-N, 1.71˚; SA-F, 5.88˚; SA-E, -1.70˚; S-ROM, 7.58˚; C2-SVA, 21.97 mm; CA-N, 9.83˚; CA-F, 11.34˚; CA-E, 3.01˚; C-ROM, 8.31˚; T1S-CA, 29.99˚) (Fig 5A). Globally, C7-SVA was -24.24 mm and the TPA was 8.30˚ (Fig 5B). PECF was performed and a ruptured disc was removed. Postoperative T2-weighted MRI showed decompression at left C5-6 and improvements in neck and arm pain (NDI, 5/50; Neck-NRS, 0/10; Arm-NRS, 0/10). X-rays taken 1 year after the operation showed that the cervical and segmental curvatures became lordotic (SA-N, -3.3˚; SA-F, 9.31˚; SA-E, -3.62˚; S-ROM, 12.93˚; CA-N, -11.57˚; CA-F, 12.64˚;

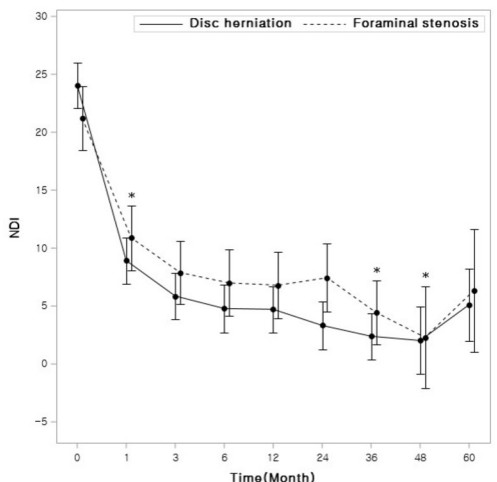

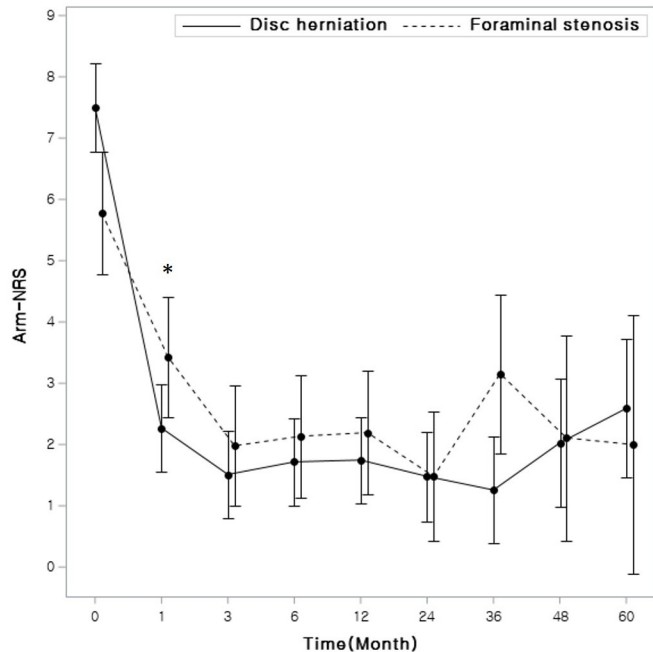

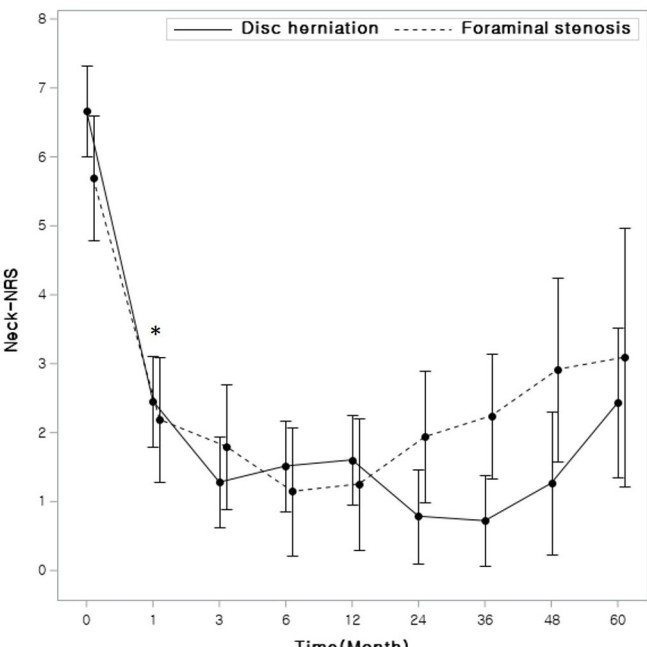

**Fig 2. Adjusted mean values from the mixed-effects model for clinical outcomes.** The Neck Disability Index (NDI) and numerical rating scale from 0 (no pain) to 10 (most severe pain) for neck pain (Neck-NRS) and arm pain (Arm-NRS) are presented as line graphs. Time 0 represents the preoperative score. An asterisk represents a statistically significant difference starting at 3 months after the operation.

CA-E, -35.59˚; C-ROM, 48.23˚; T1S-CA, 10.04˚) (Fig 5C). Globally, C7-SVA was 1.07 mm and the TPA was 7.92˚ (Fig 5D).

## Discussion

The purpose of this study was to show kinematics in patients with DH and with FS after PECF. The clinical improvements after PECF were not significantly different between groups.

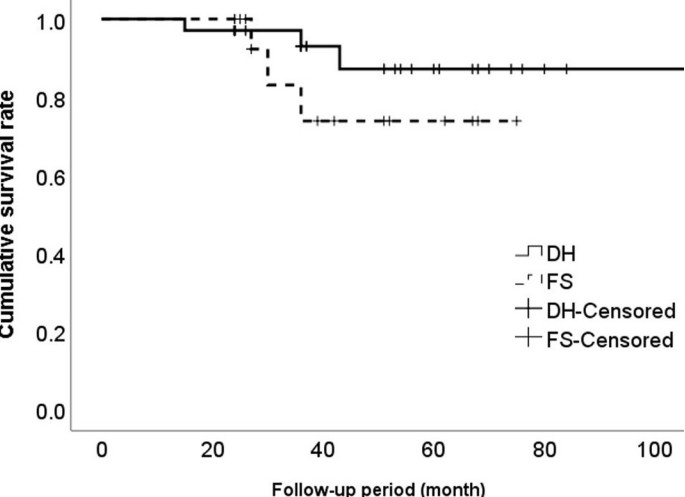

**Fig 3. Event-free survival plot.** Events and censors are represented with line graphs. Three events occurred in each group, and the event-free survival rates were 91% for disc herniation patients and 83% for foraminal stenosis patients during the follow-up period, without a significant difference between groups (P = 0.29).

Radiological parameters were evaluated not just locally, but also regionally and globally. We observed several significant changes after surgery. Locally, SA-N and SA-E became more lordotic after surgery, while disc height did not change. Although degenerative changes occurred at postoperative 2 years, further degenerative changes were not observed thereafter. Regionally, noticeable changes were a decreased T1S-CA mismatch and increased C-ROM. Globally, C7-SVA changed toward a neutral posture (Fig 5).

## Radiological changes after posterior foraminotomy

Jagannathan et al. analyzed the segmental and cervical angles after posterior open cervical foraminotomy, and observed a loss of cervical lordosis in 20% of the patients (30/162), one-third of whom had symptoms [10]. Regardless of this shortcoming, overall, posterior cervical foraminotomy has been recognized as a valid surgical procedure for patients with radiculopathy and showed a similar reoperation rate as anterior cervical discectomy and fusion [6, 8, 10, 27]. Recently, PECF has emerged as an alternative to microscopic surgery and showed comparable clinical outcomes in a randomized controlled trial and systematic review [5, 17]. The primary advantage of PECF is its minimally invasive nature thanks to the high magnification and illumination [19]. Conseque. These advantages were reflected by improved cervical lordosis after PECF, even in patients with cervical hypolordosis [14, 15].

Therefore, we sought to explore whether the advantages of PECF would be valid for patients with FS, because a larger foraminotomy is necessary in patients with FS than in patients with DH [9, 10, 19]. Patients with FS may be more likely to experience cervical kinematic changes and degeneration than patients with DH. The present study was planned to address this point, and showed that the pathology did not influence radiologic ntly, the size of foraminotomy and injuries to the posterior structure could both be minimized [19] al and clinical outcomes. Radiological parameters were assessed from local, regional, and global perspectives. Locally, segmental kinematics were well-maintained: disc height was preserved, the segmental angle became lordotic, and the segmental ROM was well-maintained throughout the follow-up period. Thus, PECF did not have a significant influence on segmental kinematics. Although PECF is not a major corrective surgery, it has gained interest for its indirect effect of pain relief

**Table 3. Radiological outcomes: Adjusted mean (standard deviation) [*].**

| | | Pre | 3 mo | 6 mo | 1 yr | 2 yr | 3 yr | 4 yr | 5 yr |
|---|---|---|---|---|---|---|---|---|---|
| SA-N | DH | 0.9 (5.5) | -3.2 (4.1) | -3.4 (4.1) | -3.2 (3.6) | -4.2 (4.5) | -2.8 (2.6) | -3.1 (3.2) | -2.5 (1.7) |
| | FS | -0.4 (3.4) | -3.3 (3) | -3.5 (3.1) | -2.7 (3.1) | -3.9 (3) | -0.7 (3.7) | -0.8 (1.6) | -0.7 (0.2) |
| SA-F | DH | 7.2 (6.3) | 6.6 (6.2) | 6.3 (5.8) | 6.9 (5.6) | 4.1 (5.5) | 7.7 (7) | 4.8 (6.9) | 8 (5.4) |
| | FS | 6.2 (5.7) | 3.4 (4.6) | 2.4 (4.5) | 4.3 (5.3) | 2 (6.2) | 5.9 (4) | 1.2 (7.6) | 5.7 (6.3) |
| SA-E | DH | -3.1 (5.6) | -6.6 (4.2) | -5 (4.9) | -6.1 (4.7) | -7.4 (4.4) | -6 (3.8) | -6.8 (4.1) | -6.2 (3.1) |
| | FS | -2.3 (5.5) | -5.3 (4.3) | -7.1 (3.6) | -5.1 (3.3) | -6.2 (4.1) | -5.2 (3.5) | -5.9 (3.2) | -2.1 (1) |
| S-ROM | DH | 10.3 (6.7) | 13.2 (5.9) | 11.3 (7.2) | 13 (5.5) | 11.5 (5.7) | 13.8 (4.7) | 10.9 (4.1) | 12.1 (8.5) |
| | FS | 8.4 (6.3) | 8.8 (2.1) | 9.5 (6.2) | 9.4 (5.3) | 8.2 (8.1) | 11.1 (5.1) | 7.2 (8.6) | 7.8 (7.3) |
| CDI | DH | 1.1 (1.2) | 1.1 (1.2) | 1.2 (1.1) | 1.2 (1.1) | 1.4 (1.6) | 1.3 (1.2) | 1.5 (1.2) | 1.3 (0.9) |
| | FS | 1.6 (1.4) | 1.5 (1.5) | 1.6 (1.5) | 1.8 (1.6) | 1.9 (1.6) | 2.4 (1.8) | 2.6 (2.5) | 3 (2) |
| aH | DH | 3.1 (1.2) | 3.3 (0.8) | 3.6 (0.9) | 2.9 (1.4) | 3.5 (0.9) | 3.5 (0.8) | 3.5 (0.6) | 3.3 (0.8) |
| | FS | 2.9 (1.5) | 3.3 (0.9) | 3.4 (1.1) | 2.7 (1.4) | 3.5 (0.7) | 2.8 (0.5) | 2.6 (0.5) | 3.3 (0.5) |
| pH | DH | 2.6 (0.9) | 2.7 (0.7) | 2.8 (0.7) | 2.6 (0.7) | 2.5 (0.7) | 3.1 (0.8) | 2.7 (0.6) | 2.8 (0.6) |
| | FS | 2.4 (1.3) | 2.5 (0.7) | 2.4 (0.7) | 2.2 (0.7) | 2.5 (0.4) | 2.3 (0.7) | 1.5 (0.3) | 2.7 (0.8) |
| C27-SVA | DH | 21.7 (11.7) | 20.6 (8.9) | 18.5 (8.9) | 17.3 (7.4) | 17 (11.5) | 16.3 (8.5) | 16.5 (3.3) | 24 (11) |
| | FS | 29.4 (18.7) | 21.7 (11.3) | 24.9 (16.4) | 20.9 (10.6) | 25.7 (12.4) | 26 (16.2) | 24.5 (17.9) | 31.8 (17.3) |
| T1S | DH | 18.7 (6.5) | 22.4 (4.9) | 21.3 (5.7) | 20.5 (4.8) | 21.7 (4.2) | 20 (6.3) | 20.2 (3.6) | 20.7 (6.9) |
| | FS | 24.6 (8.3) | 25.5 (5.3) | 27.1 (7.7) | 24.8 (6.3) | 26.6 (6.7) | 24.5 (3.4) | 23.7 (7.3) | 23.3 (5.5) |
| T1S-CA | DH | 16.3 (9.1) | 13.2 (6.7) | 12.8 (6.5) | 13.1 (7.6) | 11.5 (6.2) | 12.1 (7.9) | 12.2 (5.9) | 14.9 (7.5) |
| | FS | 16.5 (15.3) | 8.5 (8.4) | 9.8 (11.2) | 9.8 (8.7) | 12.5 (4.3) | 9.4 (6.4) | 12.1 (6.5) | 18.2 (5.7) |
| CA-N | DH | -2.4 (10.3) | -9.2 (8.2) | -8.5 (10.1) | -7.4 (8.4) | -10.2 (8.5) | -7.9 (9.2) | -8 (6.3) | -5.8 (8.8) |
| | FS | -8.1 (14.2) | -17 (9.1) | -17.3 (8.1) | -15.1 (11.2) | -14 (9) | -15.1 (8) | -9.1 (3.7) | -5 (7.1) |
| CA-F | DH | 20.1 (12.6) | 20.6 (10.8) | 20.4 (11.8) | 23.3 (10.3) | 23.8 (7.8) | 22.9 (10.9) | 20 (6.2) | 19.5 (15.6) |
| | FS | 18.3 (11.8) | 14 (15.1) | 11.7 (15.1) | 13.5 (15.3) | 19.4 (12.9) | 17.5 (13.8) | 25.1 (8.3) | 22.4 (3.3) |
| CA-E | DH | -15 (12.6) | -28.3 (8.6) | -24.9 (9.2) | -24.9 (11.3) | -30.2 (9.2) | -26.1 (13.8) | -29.1 (9.7) | -23.9 (12.8) |
| | FS | -16.2 (20.3) | -30.5 (8.1) | -30 (6.3) | -28 (7.2) | -30.3 (4.9) | -29.7 (11) | -26.6 (4.6) | -23.9 (8) |
| C-ROM | DH | 35.1 (20.5) | 48.8 (13.5) | 45.3 (13.8) | 48.1 (12.8) | 54 (12.6) | 49.1 (19.1) | 49.1 (14) | 43.3 (20.9) |
| | FS | 34.5 (22.1) | 44.5 (15.3) | 41.7 (15.1) | 41.5 (15.1) | 49.7 (12.7) | 47.2 (20.7) | 50.5 (4.2) | 46.3 (8.8) |
| C7-SVA | DH | -16.6 (21.4) | 1.9 (24.7) | -8.4 (22.5) | -6.4 (23) | -2.3 (30.2) | -1.3 (29.9) | -6.9 (18.2) | -17.1 (26.3) |
| | FS | -6.4 (21.8) | 3.6 (26.1) | 16.9 (32.1) | 8 (30.2) | 10.6 (42.4) | 3.6 (30.9) | 10.4 (37.1) | 10.8 (23.1) |
| TPA | DH | 15.5 (5.9) | 17.3 (8.6) | 15.2 (5.1) | 15.4 (7) | 15.8 (8.5) | 14.4 (7.3) | 16.7 (6) | 13.2 (6.1) |
| | FS | 15.5 (3.6) | 17 (4.9) | 18.7 (5.1) | 21.6 (5) | 19.7 (6.9) | 17.4 (4.5) | 19 (4) | 18.7 (1.9) |

[*]Adjusted mean by age, sex, body mass index, occupational activity, preoperative clinical outcome were calculated from the mixed effect model

[†]Adjusted mean for time (95% confidence interval)

[‡]Statistically significant at the 0.05 level.

Abbreviations: C7-SVA, C7 sagittal vertical axis; TPA, T1 pelvic angle; C27-SVA, C2 to C7 sagittal vertical axis; T1S, T1 slope; CA-N, cervical neural curvature from C2-7; CA-F, cervical flexion curvature; CA-E, cervical extension curvature; T1S-CA, difference between T1 slope and CA-N; C-ROM, cervical range of motion; SA-N, index level segmental neutral angle; SA-F, segmental flexion angle; SA-E, segmental extension angle; S-ROM, segmental range of motion; aH, anterior disc height; pH, posterior disc height; CDI, cervical degenerative index.

on the regional and global scale [28–30]. Regionally, the patients were able to achieve a more lordotic cervical posture, and extend and move their neck better than before surgery. The gap between T1 slope and cervical posture was narrowed by around 5°. Consequently, the neck moved closer to the gravity line. Overall, patients' postures became more comfortable after surgery [31–33]. The changes of local, regional, and global curvature may be speculated like followings [31, 33–35]. Patient could extend neck more freely without pain and this may explain

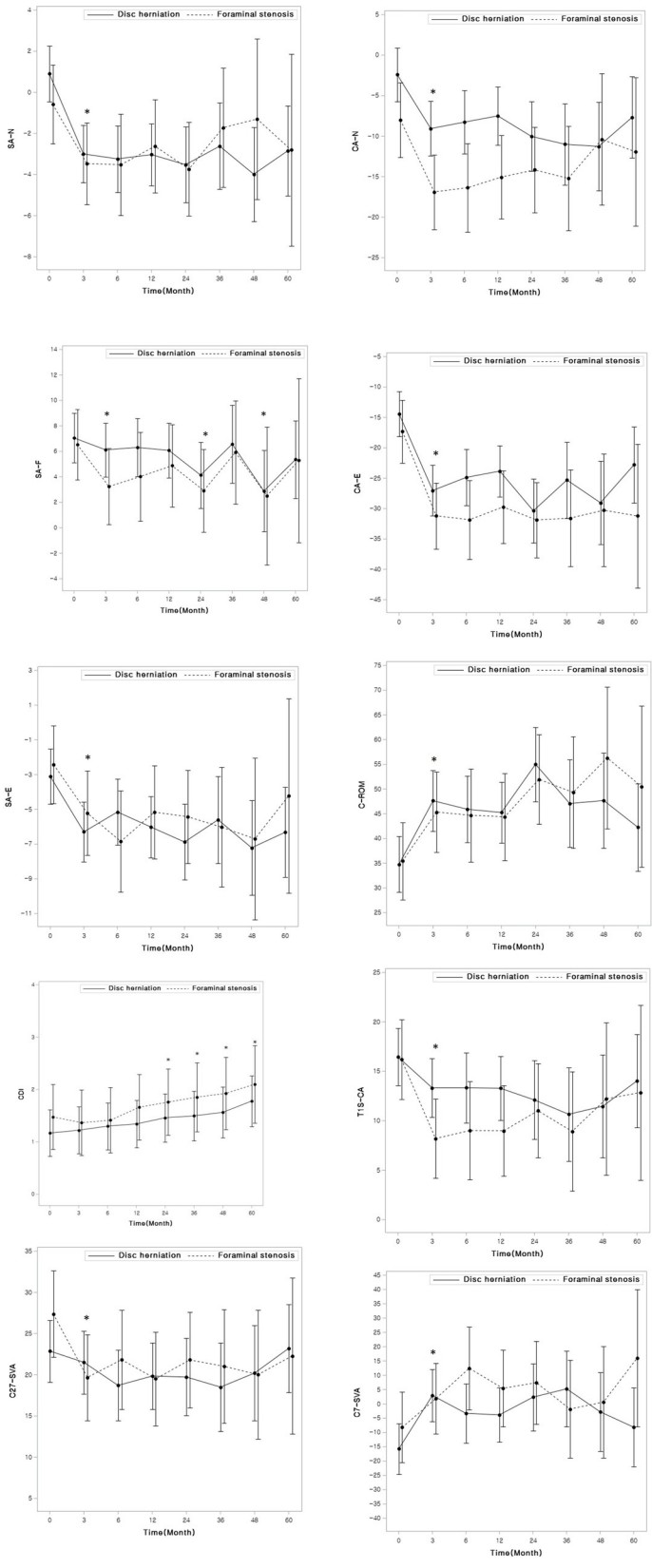

**Fig 4. Radiological outcomes.** Adjusted mean values and standard deviations are plotted on line graphs. Statistically significant changes (P < 0.01) starting at 3 months after the operation are marked with asterisks. a. Index level segmental neutral angle (SA-N). b. Segmental flexion angle (SA-F). c. Segmental extension angle (SA-E). d. Cervical degenerative index (CDI). E. C2 to C7 sagittal vertical axis (C27-SVA). F. Cervical neural curvature from C2-7 (CA-N). g. Cervical extension curvature (CA-E). h. Cervical range of motion (C-ROM). i. Difference between T1 slope and CA-N (T1S-CA). j. C7 sagittal vertical axis (C7-SVA).

improved regional cervical curvature. Although there were no statistically significant changes in anterior and posterior disc height (aH and pH, Table 3), the heights were not the same before and after surgery and those minimal change by improved cervical lordosis might have changed local curvature. Although causal relationships could not be inferred based on this study, patients with decreased neck pain were more likely to be able to take an upright posture and this may influence on global curvature.

We suggest that the indication of PECT would be the most important factor for satisfactory outcomes. As mentioned, PECF was indicated for patients 1) with single-level unilateral radiculopathy due to cervical DH or FS, 2) a positive Spurling's test, 3) disc space narrowing not more than 50%, [20]. Although the shape of cervical curvature was not specified in this study, cervical kyphosis more than 10 degrees was not a contraindication of PECF, if the curvature was not a structural change reflected by such as decreased disc height more than 50%, foraminal arthrosis or spur change at endplate [15, 16, 19]. The purpose of PECF was to relieve pain and the change of curvature was a secondary change after relief of pain. Therefore, PECF is usually indicated for patient with radiculopathy, mild cervical degeneration, and functional curvature change and it should not be considered to correct cervical curvature when the curvature change was structural one.

## Reoperation

The present study showed that adverse events occurred in 6 patients, and secondary surgery was done for 1 patient in the FS group (6%) at the index level. Lubelski et al. reported a reoperation rate of 6.4% at the index level after posterior open cervical foraminotomy during 2 years of postoperative follow-up [27]. Although PECF is a minimally invasive surgical technique, it is not a regenerative treatment and degeneration naturally progressed at postoperative 2 years as showed in this study. Nonetheless, further degeneration did not occur thereafter, and a further comparative study with a control group is necessary to determine whether PECF hastened the progression of degeneration [20, 36]. In addition, the advantages of PECF compared to open foraminotomy need to be verified by a comparative study. Although many questions remain to be answered, the present study showed an event-free rate of 80%–90% after PECF, and this information could be helpful in a shared decision-making process.

## Limitations

The present study tried to compare the kinematics between DH and FS, but was underpowered due to its small sample size. A larger number of patients would be necessary to overcome type I or II error. Second, kinematic changes were assessed by static flexion and extension X-rays, which were not capable of showing kinematics between those positions [37]. Third, surgical injury of the facet joint and musculature by PECF may also indirectly counteract the positive effects on cervical curvature that are obtained by the alleviation of pain [14]. In addition, the size of foraminotomy was not measured in computed tomography or magnetic resonance imaging, which was not routinely ordered without symptom. Therefore, this study was not to evaluate kinematic change according to the size of foraminotomy. Long-term follow-up

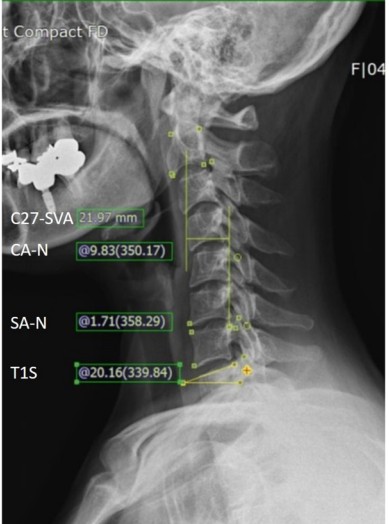

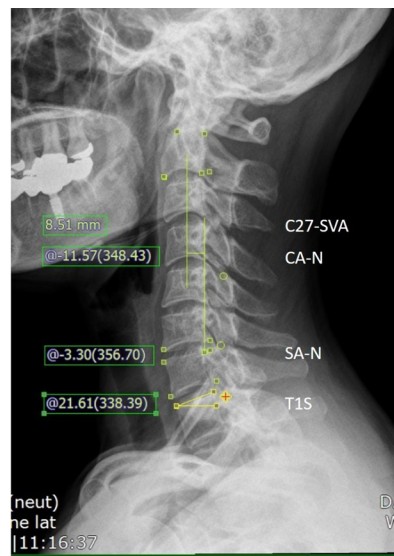

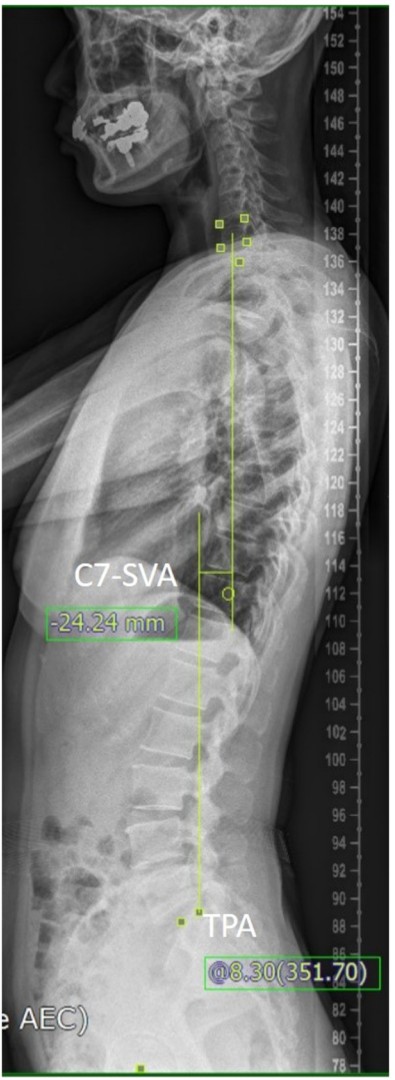

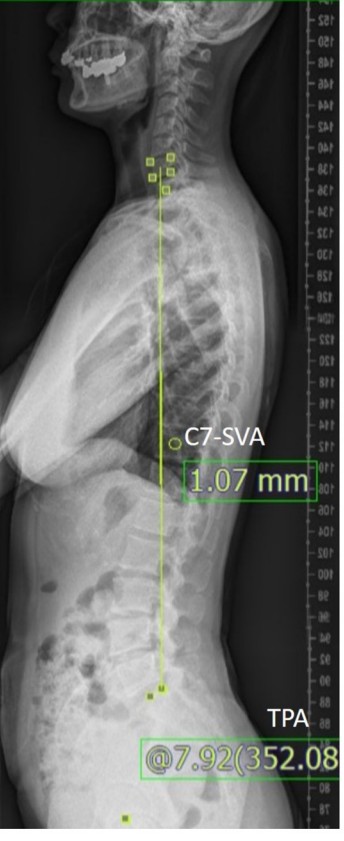

**Fig 5. Case.** Preoperative standing lateral neutral cervical-spine X-ray (a). Radiological parameters were measured on 150% magnified images using measuring tools in the institution's picture archiving and communication system (Marosis, version 5483, Infinitt Healthcare, Seoul, Korea) and measurements were marked in radiographs. Preoperative whole-spine standing lateral X-ray (b). Postoperative 1 year, standing lateral neutral cervical-spine X-ray (c) and whole-spine standing lateral X-ray (d). Abbreviations: SA-N, segmental neutral angle; C27-SVA, C2 to C7 sagittal vertical axis; T1S, T1 slope; CA-N, cervical neural curvature from C2-7; C7-SVA, C7 sagittal vertical axis; TPA, T1 pelvic angle.

observations of a large number of patients are required to identify the trade-off between the natural restoration of curvature and the aggravation of curvature due to surgical trauma [14]. Fourth, this study did not compare PECF and open foraminotomy. In addition, the advantages of PECF compared to conventional foraminotomy cannot be evaluated based on the current results. A prospective cohort study or randomized controlled trial would be necessary to compare kinematics between these procedures. Regardless of those shortcomings, the present study at least showed that the underlying pathology may not worsen cervical kinematics if the surgical insult is minimized, as the patients were able to take a comfortable posture involving economical movements after surgery due to decreased pain and muscle spasms [15, 38]. When indicated, PECF may be an alternative surgical option for motion preservation, even for patients with FS [21].

## Conclusions

The clinical and radiological outcomes after PECF were not significantly different between patients with disc herniation and patients with foraminal stenosis. To have the best outcome, the indication of PECF should be kept in mind. PECT is usually indicated for patient with radiculopathy, mild cervical degeneration, and functional curvature change. It may not be simple to tell functional from structural change of curvature, but sings of moderate to severe degeneration, such as decreased disc height more than 50%, facet arthrosis and spur change may indicate structural change. In such case, a secondary change of cervical curvature would not occur, and this should be considered in deciding surgical techniques. These findings will be informative for surgeons and patients during the shared decision-making process.

## Supporting information

**S1 Data.**
(SAV)

## Acknowledgments

I appreciate kind consideration of [Blinded for review] for English editing.

## Author Contributions

**Conceptualization:** Seungyoon Paik, Yunhee Choi, John Min Rhee, Chi Heon Kim.

**Data curation:** Seungyoon Paik, Yunhee Choi, Chi Heon Kim.

**Formal analysis:** Seungyoon Paik, Yunhee Choi, Chi Heon Kim.

**Funding acquisition:** Chi Heon Kim.

**Investigation:** Seungyoon Paik, Seung Heon Yang, Chi Heon Kim.

**Methodology:** Seungyoon Paik, Yunhee Choi, Chun Kee Chung, Chi Heon Kim.

**Project administration:** Seungyoon Paik, Young Il Won.

**Resources:** Yunhee Choi, Chi Heon Kim.

**Software:** Yunhee Choi, Chi Heon Kim.

**Supervision:** Chun Kee Chung, Chang-Hyun Lee.

**Validation:** Yunhee Choi, John Min Rhee, Kyoung-Tae Kim.

**Visualization:** Yunhee Choi.

**Writing – original draft:** Seungyoon Paik, Yunhee Choi, Chi Heon Kim.

**Writing – review & editing:** Sung Bae Park, John Min Rhee, Kyoung-Tae Kim.

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
