## [Decision Letter · Decision Letter 0]

26 Jan 2023

PONE-D-22-01741Cervical kinematic change after posterior full-endoscopic cervical foraminotomy for disc herniation or foraminal stenosisPLOS ONE

Dear Dr. Kim,

Thank you for submitting your manuscript to PLOS ONE. After careful consideration, we feel that it has merit but does not fully meet PLOS ONE’s publication criteria as it currently stands. Therefore, we invite you to submit a revised version of the manuscript that addresses the points raised during the review process.

We look forward to receiving your revised manuscript.

Kind regards,

Thamer Hamdan, PhD

Academic Editor

PLOS ONE

Journal Requirements:

"This work was supported by the New Faculty Startup Fund from Seoul National University. This study was supported by grant no. 04-2021-0540 from Seoul National University Hospital research fund."

"I have read the journal's policy and the authors of this manuscript have the following competing interests: The corresponding author (CHK) is a consultant of RIWOspine GmBH. All the other authors declare that they have no conflicts of interest concerning the materials/methods used in this study or the findings described in this paper. No benefits in any form have been or will be received from any commercial party related directly or indirectly to the subject of this manuscript."

Reviewers' comments:

Reviewer's Responses to Questions

**Comments to the Author**

1. Is the manuscript technically sound, and do the data support the conclusions?

Reviewer #1: Yes

Reviewer #2: Yes

Reviewer #3: Yes

2. Has the statistical analysis been performed appropriately and rigorously? 

Reviewer #1: Yes

Reviewer #2: Yes

Reviewer #3: Yes

3. Have the authors made all data underlying the findings in their manuscript fully available?

Reviewer #1: Yes

Reviewer #2: Yes

Reviewer #3: Yes

4. Is the manuscript presented in an intelligible fashion and written in standard English?

Reviewer #1: Yes

Reviewer #2: Yes

Reviewer #3: Yes

5. Review Comments to the Author

Reviewer #1: Dear authors, I congratulate your work on this subject.

Line 51, 359, 374: please explain in the discussion, why cervical curvature become more lordotic while disc height not changed? (is there other explanation than due to pain relieve, line 412 ?)

Reviewer #2: Thank you for your submitting this research and hopefully you can overcome the limitations

Reviewer #3: This is an excellent and well-made article. The topic is of interest in the neverending debate on what is the best approach for foraminal stenosis or intraforaminal disc herniation. Unfortunately, the sample size is relatively small, but the author accounts for this in the limitation section.

I will add some minor revisions to the discussion part:

1) Please add a paragraph specifying the best indications for PECF. More specifically, what are the best candidates from and anatomical (i.e., neck lordosis, shape, etc.) and from type of pathlogy point of view (i.e., foraminal stenosis, disc herniation). It is relatively unclear to the readers and is not specified well enough in the text.

2) Please add a paragraph explaining to the readers what is the reasoning process that the spine surgeon should keep in mind in patient selection after considering the results from your analysis.

6. PLOS authors have the option to publish the peer review history of their article (what does this mean?). If published, this will include your full peer review and any attached files.

Reviewer #1: **Yes: **Raed J. Chasib

Reviewer #2: No

Reviewer #3: **Yes: **Enrico Giordan

---

## [Author Response · Author response to Decision Letter 0]

29 Jan 2023

Response to Reviewers

PONE-D-22-01741

Cervical kinematic change after posterior full-endoscopic cervical foraminotomy for disc herniation or foraminal stenosis

PLOS ONE

Reviewers' comments:

Reviewer #1: Dear authors, I congratulate your work on this subject.

Line 51, 359, 374: please explain in the discussion, why cervical curvature become more lordotic while disc height not changed? (is there other explanation than due to pain relieve, line 412 ?)

Answer. 

I appreciate encouraging comment. We discussed the comment in the discussion section as follows (line 395)

The changes of local, regional, and global curvature may be speculated like followings. [31,33-35] Patient could extend neck more freely without pain and this may explain improved regional cervical curvature. Although there were no statistically significant changes in anterior and posterior disc height (aH and pH, Table 3), the heights were not the same before and after surgery and those minimal change by improved cervical lordosis might have changed local curvature. Although causal relationships could not be inferred based on this study, patients with decreased neck pain were more likely to be able to take an upright posture and this may influence on global curvature.

Reviewer #2: Thank you for your submitting this research and hopefully you can overcome the limitations. 

Answer. 

I appreciate encouraging comment. We are planning new projects regarding the issues in the limitation section. I hope to submit the new projects in PLOS one. Thank you very much.

Reviewer #3: This is an excellent and well-made article. The topic is of interest in the never ending debate on what is the best approach for foraminal stenosis or intraforaminal disc herniation. Unfortunately, the sample size is relatively small, but the author accounts for this in the limitation section.

I will add some minor revisions to the discussion part:

1) Please add a paragraph specifying the best indications for PECF. More specifically, what are the best candidates from and anatomical (i.e., neck lordosis, shape, etc.) and from type of pathlogy point of view (i.e., foraminal stenosis, disc herniation). It is relatively unclear to the readers and is not specified well enough in the text.

Answer.

I appreciate the encouraging comment. We clarified the indication as follows in the discussion section. Thank you. 

Line 403

We suggest that the indication of PECT would be the most important factor for satisfactory outcomes. As mentioned, PECF was indicated for patients 1) with single-level unilateral radiculopathy due to cervical DH or FS, 2) a positive Spurling’s test, 3) disc space narrowing not more than 50%,[20]. Although the shape of cervical curvature was not specified in this study, cervical kyphosis more than 10 degrees was not a contraindication of PECF, if the curvature was not a structural change reflected by such as decreased disc height more than 50%, foraminal arthrosis or spur change at endplate. [15,16,19] The purpose of PECF was to relieve pain and the change of curvature was a secondary change after relief of pain. Therefore, PECF is usually indicated for patient with radiculopathy, mild cervical degeneration, and functional curvature change and it should not be considered to correct cervical curvature when the curvature change was structural one.

2) Please add a paragraph explaining to the readers what is the reasoning process that the spine surgeon should keep in mind in patient selection after considering the results from your analysis.

Answer. 

I appreciate the comment. Indication of PECT would be the most important one for readers. I emphasized it in the conclusion section as follows. Thank you. 

Line 452. 

To have the best outcome, the indication of PECF should be kept in mind. PECT is usually indicated for patient with radiculopathy, mild cervical degeneration, and functional curvature change. It may not be simple to tell functional from structural change of curvature, but sings of moderate to severe degeneration, such as decreased disc height more than 50%, facet arthrosis and spur change may indicate structural change. In such case, a secondary change of cervical curvature would not occur, and this should be considered in deciding surgical techniques.

---

## [Editor Report · Decision Letter 1]

5 Feb 2023

Cervical kinematic change after posterior full-endoscopic cervical foraminotomy for disc herniation or foraminal stenosis

PONE-D-22-01741R1

Dear Dr. Kim,

We’re pleased to inform you that your manuscript has been judged scientifically suitable for publication and will be formally accepted for publication once it meets all outstanding technical requirements.

Kind regards,

Thamer Hamdan, PhD

Academic Editor

PLOS ONE
---

## [Editor Report · Acceptance letter]

9 Feb 2023

PONE-D-22-01741R1 

Cervical kinematic change after posterior full-endoscopic cervical foraminotomy for disc herniation or foraminal stenosis 

Dear Dr. Kim:

I'm pleased to inform you that your manuscript has been deemed suitable for publication in PLOS ONE. Congratulations! Your manuscript is now with our production department. 

Kind regards, 

on behalf of

Professor Thamer Hamdan 

Academic Editor

PLOS ONE